# Neonatal brain injury influences structural connectivity and childhood functional outcomes

**Alice Ramirez[1], Shabnam Peyvandi[1], Stephany Cox[1], Dawn Gano[1,2], Duan Xu[3], Olga Tymofiyeva[3], Patrick S. McQuillen[1,2]***

**1** Department of Pediatrics, University of California, San Francisco, San Francisco, California, United States of America, **2** Department of Neurology, University of California, San Francisco, San Francisco, California, United States of America, **3** Department of Radiology, University of California, San Francisco, San Francisco, California, United States of America

* patrick.mcquillen@ucsf.edu

## Abstract

Neonatal brain injury may impact brain development and lead to lifelong functional impairments. Hypoxic-ischemic encephalopathy (HIE) and congenital heart disease (CHD) are two common causes of neonatal brain injury differing in timing and mechanism. Maturation of whole-brain neural networks can be quantified during development using diffusion magnetic resonance imaging (dMRI) in combination with graph theory metrics. DMRI of 35 subjects with CHD and 62 subjects with HIE were compared to understand differences in the effects of HIE and CHD on the development of network topological parameters and functional outcomes. CHD newborns had worse 12–18 month language (P<0.01) and 30 month cognitive (P<0.01), language (P = 0.05), motor outcomes (P = 0.01). Global efficiency, a metric of brain integration, was lower in CHD (P = 0.03) than in HIE, but transitivity, modularity and small-worldness were similar. After controlling for clinical factors known to affect neurodevelopmental outcomes, we observed that global efficiency was highly associated with 30 month motor outcomes (P = 0.02) in both groups. To explore neural correlates of adverse language outcomes in CHD, we used hypothesis-based and data-driven approaches to identify pathways with altered structural connectivity. We found that connectivity strength in the superior longitudinal fasciculus (SLF) tract 2 was inversely associated with expressive language. After false discovery rate correction, a whole connectome edge analysis identified 18 pathways that were hypoconnected in the CHD cohort as compared to HIE. In sum, our study shows that neonatal structural connectivity predicts early motor development after HIE or in subjects with CHD, and regional SLF connectivity is associated with language outcomes. Further research is needed to determine if and how brain networks change over time and whether those changes represent recovery or ongoing dysfunction. This knowledge will directly inform strategies to optimize neurologic functional outcomes after neonatal brain injury.

**Data Availability Statement:** Although the final dataset will be stripped of identifiers prior to release for sharing, we believe that there remains the possibility of deductive disclosure of subjects

with unusual characteristics. Adhering to CHR mandated HIPPA protections, we will make the data and associated documentation available under a data-sharing agreement or institutional data usage agreement that provides for the following: (1) a commitment to using the data only for research purposes and not to identify any individual participant; (2) a commitment to securing the data using appropriate computer technology; and (3) a commitment to destroying or returning the data after analyses are completed. Requests to establish the necessary data use agreements for access to the primary dataset can be sent via email to the UCSF Office of Sponsored Research (industrycontracts@ucsf.edu).

**Funding:** We would further like to emphasize that the funders [PSM: P01 NS082330 (https://www. ninds.nih.gov/) and SP: K23 NS099422 (https:// www.ninds.nih.gov/)] had no role in study design, data collection and analysis, decision to publish, or preparation of the manuscript.

**Competing interests:** I have read the journal's policy and the authors of this manuscript have the following competing interests: Dawn Gano received grant funding from the following organizations within a 5 year period of this article: - Cerebral Palsy Alliance - UCSF preterm birth initiative funded by Marc & Lynne Benioff.

## Introduction

Hypoxic-ischemic encephalopathy (HIE) and severe congenital heart disease (CHD) are two of the most common sources of term neonatal brain injury [1], although injury timing and mechanism is markedly different between the two conditions. Studies of neurodevelopmental outcome in children with CHD [2] or HIE [3] in comparison with normal cohorts have shown morbidities affecting many neurocognitive domains—intelligence, visuomotor skills, memory, language, peer relationships, emotion, and social conduct [4, 5]. Severe CHD includes conditions with malformations of heart development (e.g. transposition of the great arteries—TGA, hypoplastic left heart syndrome—HLHS) that decrease fetal brain oxygen and substrate delivery [6] throughout gestation requiring life-saving surgery shortly after birth. HIE is a form of neonatal encephalopathy resulting from impaired brain oxygen delivery leading to brain injury of variable severity [7]. In contrast to CHD, HIE brain injury occurs in the days/weeks leading up to birth or within hours after birth from either sub-acute (e.g. placental insufficiency) or acute (e.g. placental abruption, uterine rupture) mechanisms disrupting brain oxygen delivery [7]. Acute injury in HIE occurs after a period of normal fetal brain development. Chronic disruption of brain oxygen delivery and brain growth in the third trimester in CHD may disrupt the formation and refinement of brain networks; whereas in HIE, acute or subacute brain injury damages or destroys brain connections that hitherto developed normally. Comparing these two cohorts with markedly different timing and mechanism of neonatal brain injury provides a 'natural experiment' to understand the consequences on different brain injuries on brain development.

Human brain development is characterized by the formation of precise neuronal circuits through a process of activity dependent refinement from initially imprecise connections occurring from the third trimester of pregnancy into infancy [8]. Non-invasive visualization of brain white matter structure has become possible through advances in diffusion magnetic resonance imaging (dMRI). White matter tracts can be derived from dMRI, and the patterns of white matter tract connections, i.e. structural connectivity, can be quantitatively described using topological graph theory metrics [9]. This application of graph theory to model neural connectivity within the brain is called "connectomics", and the technique has been widely applied within neuropsychiatric research [10]. Analysis of brain structural connectomes in healthy fetuses and neonates after uncomplicated term delivery has afforded increasingly detailed description of normal brain network development [11]. Structural connectivity has been combined with detailed neuropsychological testing to make brain behavior correlations that provide insight into disease [12–14].

Differences in structural connectivity have been noted in CHD newborns before surgery [15, 16] and in newborns after HIE [17, 18]. In the present study, we sought to determine if neonatal structural connectivity was different between term newborns with HIE and CHD, and if so, to analyze the association of network topological parameters with later neurodevelopmental outcomes. By doing so, we can gain insight into mechanisms and significance of the disruption to normal brain development by specific forms of neonatal brain injury. We hypothesize that impaired fetal brain oxygen delivery in CHD results in more widespread disruption of the development of structural connectivity networks compared with HIE due to effects across the third trimester. Furthermore, we expect that differences in timing and mechanisms of brain injury between HIE and CHD will be associated with selective vulnerability of distinct brain networks and these regional differences may explain specific functional disability in the two conditions.

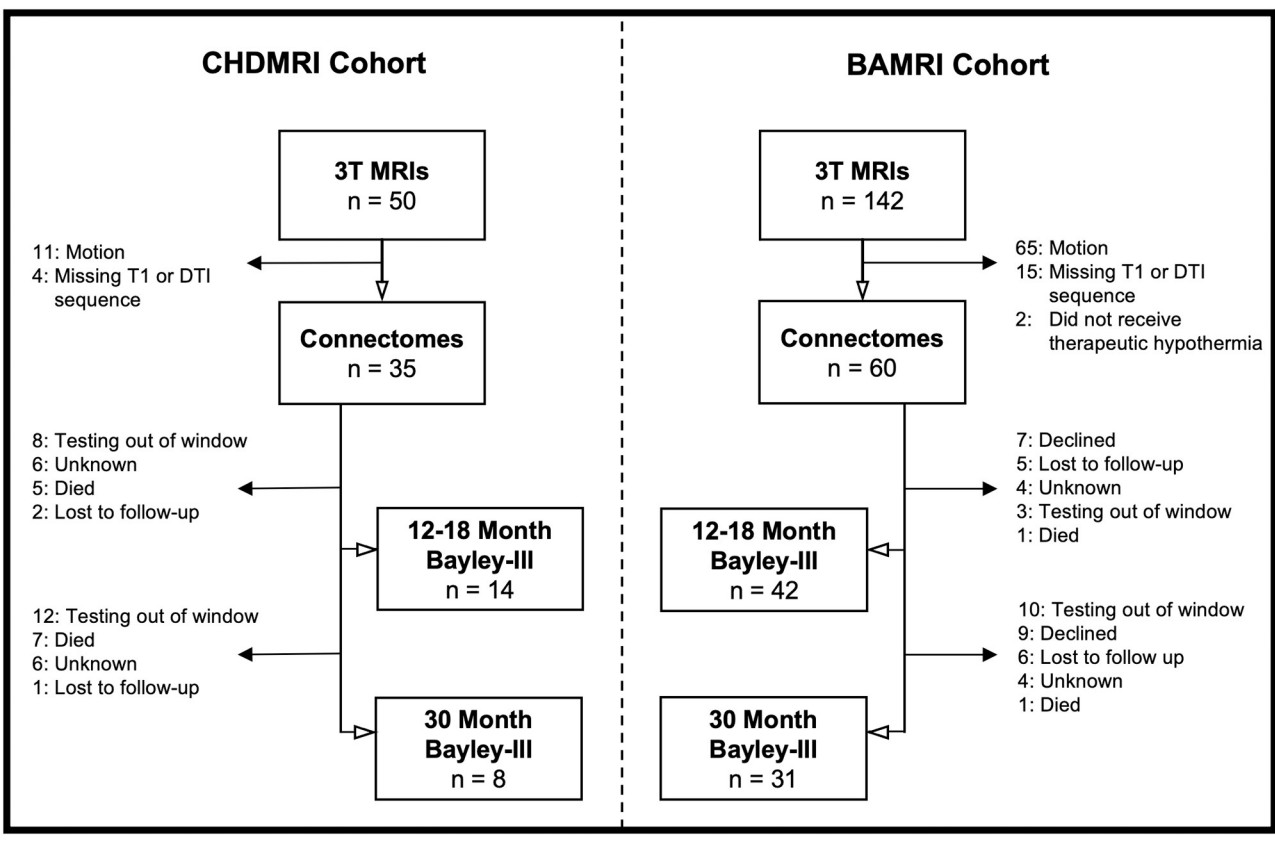

**Fig 1. Study diagram.**

## Methods

### Participants

Our study sample was derived from two single center prospective cohort studies—CHDMRI (Congenital Heart Disease MRI) and BAMRI (Birth Asphyxia MRI) (Fig 1). Neither of these studies enrolled normal newborns that could provide a contemporary control cohort.

CHDMRI is a prospective cohort study enrolling newborns with critical CHD at the University of California San Francisco (UCSF) to undergo pre- and post-operative brain MRI. Exclusion criteria includes prematurity ($< 36$ weeks) and the presence of other suspected congenital and/or genetic anomalies. Subjects were enrolled after voluntary informed consent was obtained from the parents following a protocol approved by the UCSF Committee on Human Research. We selected participants in the study who had 3T MRI imaging before surgery (2011–2020) and had either single ventricle physiology (SV) or transposition of the great arteries (TGA). Study subjects had neurodevelopmental follow up at 12–18 months and 30 months of life. Three subjects were later diagnosed with genetic anomalies (47XYY, unbalanced translocation between chromosomes 10 and 17, KBG syndrome). These subjects had similar graph metrics to the rest of their cohort and thus were included in the structural connectivity analysis, however they were excluded from neurodevelopmental analyses.

The BAMRI study is also a prospective cohort study of term neonates treated at UCSF for HIE. Subjects had moderate-severe encephalopathy presumed due to hypoxia-ischemia based on presence of one of the following: pH $<7.1$ on umbilical arterial cord gas, base deficit worse

than -10, or an Apgar score under five at five minutes of life. Newborns were excluded if there were suspected or confirmed infectious, metabolic, or congenital malformations including CHD. Subjects were enrolled after voluntary informed consent was obtained from the parents following a protocol approved by the UCSF Committee on Human Research. We selected participants who had 3T MRI imaging during the first week of life (2011–2018). Study subjects had neurodevelopmental follow up at 12–18 months and 30 months of life. In our sample, all but 2 subjects received therapeutic hypothermia, and these were excluded from analysis.

### Clinical data acquisition

Data was collected from the electronic medical record and the CHDMRI and BAMRI study datasets. Clinical data collected included sex, gestational age at birth and MRI, birth weight, and race. Peri-delivery clinical markers collected included pH, base deficit, Apgar scores, and the Score for Neonatal Acute Physiology—Perinatal Extension (SNAPPE) score. The Apgar score is routinely performed in one and five minute intervals immediately after delivery and is an assessment of birth transition with a range of 0–10. Higher scores indicate greater health. The SNAPPE score is a validated score of perinatal level of illness and has a range of 0–162, with higher scores indicating more severe illness [19]. In the HIE cohort, we also assessed presence of clinical and electrographic seizures during the first three days of life.

### Neurodevelopmental testing

Subjects were tested using the Bayley Scales of Infant & Toddler Development, Ed. 3 (Bayley-III) at 12–18 months and at 30 months of age by a psychologist blinded to clinical and neuro-imaging data. The Bayley-III assesses cognitive, language, and motor function separately. The population average and standard deviation for each domain is 100 +/- 15 points, with higher numbers indicating higher performance. The language domain is composed of both receptive and expressive language sub-scores. Sub-scores range from 0 to 20 (population average of 10).

### MRI acquisition and processing

MR images during the first week of life were acquired using a 3T General Electric EXCITE MRI scanner. MR scans included sagittal volumetric 3D spoiled gradient-echo (SPGR) T1-weighted images and axial diffusion tensor imaging (DTI) using 30 directions distributed by electromagnetic repulsion and b-value of 700 s/mm2.

Acquired injury was identified on structural and diffusion weighted MRI sequences using published scoring systems. For BAMRI subjects the degree of watershed and basal ganglia injury pattern was scored [20], and for CHDMRI subjects, the degree of white matter injury or stroke was scored [21].

Further processing was performed to create connectomes (Fig 2). An automated data rejection algorithm was used to discard motion corrupted DTI images as described previously for term neonates (24). Deterministic tractography was performed using the Diffusion Toolkit [22, 23]. Ninety cerebral regions of interest (ROIs) were determined by mapping the Neonatal UNC Infant012 Atlas to T1-SPGR images [24]. Co-registration of the diffusion and T1 images was performed using the FMRIB Linear Image Registration Tool (FLIRT), and alignment was visually inspected allowing for further rejection of motion corrupted images [25, 26]. Undirected adjacency matrices (connectomes) weighted by fractional anisotropy (FA) were constructed for each infant, wherein each ROI is connected by a set of tracts and the weight is determined via averaged FA along those tracts.

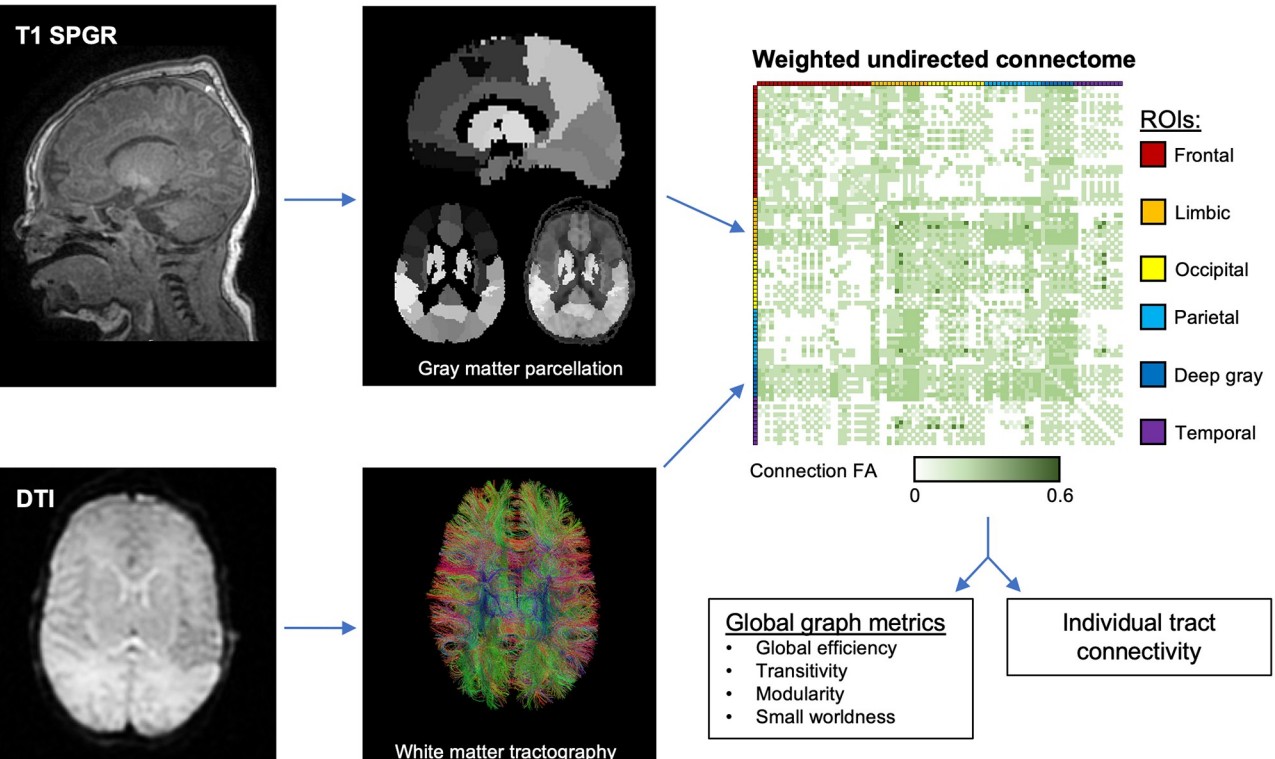

**Fig 2. MRI processing.** T1 images are registered to the neonatal UNC Infant 012 Atlas and visually inspected for motion degradation. An automated rejection algorithm inspects and eliminates corrupted directions of DTI imaging, and deterministic tractography is performed to create white matter tracts. The connectome, a weighted undirected adjacency matrix between each of the 90 cerebral regions of interests (ROIs) is created, with average fractional anisotropy (FA) along the connecting tracts used as weights. Connection patterns are described using global graph theory metrics and individual tract connectivity is assessed.

### Graph analysis

Graph parameters were then derived in Matlab using the Brain Connectivity Toolkit [9]. The following graph parameters were calculated: global efficiency, transitivity, modularity, and small-worldness. Global efficiency, a measure of network integration, is the averaged inverse of the shortest path between any two ROIs. Transitivity reflects the degree to which connections are clustered around individual ROIs, and thus is a measure of network segregation. Modularity is a measure of community structure, parceling the ROIs into non-overlapping subdivisions and calculating the degree to which connections occur within the subdivision versus between the subdivisions. When calculating modularity, subdivision size is optimized for maximal in-neighborhood connections. Small-worldness is a measure of community structure reliant on hub-spoke formation (hierarchical structuring). It is a ratio comparing clustered connections around ROIs to the overall network path lengths, as compared to randomly generated networks with the same number of ROIs and connections. Small-worldness values greater than 1, therefore, indicate that the brain has increased hierarchical organization than that of a randomly connected brain.

Because the largest difference in neurodevelopmental outcome between groups was seen in the language domain, FA-weighted connectivity strength along left-sided white matter tracts associated with language function was explored. We analyzed the superior longitudinal fasciculus (SLF), inferior longitudinal fasciculus (ILF), and arcuate fasciculus (AF).

The SLF is comprised of three main tracts: SLF 1 connects the superior parietal cortex to the superior frontal cortex and supplemental motor area, SLF 2 connects the inferior parietal superior and mid-frontal cortices, and SLF 3 connects the supramarginal gyrus to the prefrontal cortex between [27]. By mapping these tracts to the ROIs of the UNC Infant 012 Atlas, we analyzed the SLF 1 connectivity by measuring average FA along the white matter tracts connecting the superior parietal gyrus to the superior frontal gyrus (dorsal), superior frontal gyrus (medial), and supplementary motor area. Similarly, we analyzed the SLF 2 connectivity by measuring FA along the white matter tracts connecting the inferior parietal lobule to the middle frontal gyrus and superior frontal gyrus. SLF 3 connectivity was measured from the supramarginal gyrus to the middle frontal gyrus, superior frontal gyrus, inferior orbitofrontal cortex, middle orbitofrontal cortex, superior orbitofrontal cortex, opercular inferior frontal gyrus, and triangular inferior frontal gyrus.

The ILF is a white matter bundle connecting the occipital, lingual, and cuneal cortices to the anterior temporal regions [28]. By mapping these tracts to ROIs in the UNC Infant 012 Atlas, we analyzed ILF connectivity as total FA along the tracts between the following regions: fusiform gyrus to superior temporal pole, fusiform gyrus to middle temporal pole, superior occipital gyrus to superior temporal pole, middle occipital gyrus to superior temporal pole, inferior occipital gyrus to superior temporal pole, superior occipital gyrus to middle temporal pole, middle occipital gyrus to middle temporal pole, inferior occipital gyrus to middle temporal pole, lingual gyrus to middle temporal gyrus, lingual gyrus to middle temporal pole, cuneus to middle temporal gyrus, and the cuneus to middle temporal pole.

The AF connects the superior and middle temporal cortices to Brodmann areas 44, 45 and 47 [29]. Mapping to the UNC Infant 012 Atlas, we analyzed AF connectivity between the superior and middle temporal gyri to the opercular and triangular inferior frontal cortices as well as orbitofrontal cortex.

## Statistical processing

Direct statistical comparisons for continuous outcome variables were performed using non-parametric rank-sum testing. For categorical outcome variables, fisher's exact or chi-squared testing was performed. Univariate relationships between neurodevelopmental outcomes and patient characteristics including graph metrics and SLF adjacency were performed using linear regression. Multivariate linear regression models were constructed using variables that had a p value less than 0.1 in the univariate analysis. Because graph metrics are not independent, a separate model was created for each metric with a p value under 0.1 in the univariate analysis. Corrected gestational age at MRI was included a priori in each multivariate model to control for age-related developmental progression in connectivity. Gestational age at birth was colinear with corrected gestational age and was excluded. Graph metrics were log-transformed to optimize distribution normality.

In addition, connectivity between each ROI was compared between groups, controlling for gestational age at MRI and for a false discovery rate using a standard link-based controlling procedure made available for Matlab by Zalesky et al with default parameters (significance value of 0.05 and 5000 permutations) [30]. All statistical processing was performed in Matlab and StataIC 16.

## Results

### Study subjects

The MRI scans of 35 subjects with CHD and 60 subjects with HIE were included. Of the CHD cohort, 20 had SV physiology and 15 had TGA. Demographics characteristics are shown in

**Table 1. Demographics.**

|  | HIE | CHD |  |
|---|---|---|---|
|  | **n = 60** | **n = 35** |  |
|  |  | **(SV 20, TGA 15)** |  |
| **Characteristic** |  |  | **P values** |
| Male Sex–n (%) | **31 (51.7%)** | **28 (80.0%)** | **<0.01** |
| Birth Weight–kg, median (IQR) | 3.29 (2.98–3.71) | 3.24 (2.95–3.55) | 0.52 |
| Gestational Age at Birth–weeks, median (IQR) | **39.9, IQ (38.1–40.7)** | **39.1 (37.9–39.4)** | **0.01** |
| Gestational Age at MRI–weeks, median (IQR) | **40.6 (38.7–41.4)** | **39.6 (38.7–40.1)** | **0.02** |

HIE, hypoxic-ischemic encephalopathy; CHD, congenital heart disease; SV, single ventricle; TGA, transposition of the great arteries; IQR, interquartile range. P values calculated by Fischer's exact and non-parametric rank sum tests.

Table 1. The CHD cohort is notable for a younger gestational age at birth (0.8 weeks younger, p = 0.01) and at time of MRI (1 week younger, p = 0.02). The CHD cohort is more predominately male (80% v 51.7%, p < 0.01). No difference is seen in birthweight.

Forty (66.7%) HIE subjects and 22 (62.9%) CHD subjects (preoperative MRI) had normal MRIs (Table 2). The clinical characteristics at baseline did not differ between those who had normal and abnormal MRIs. Patients with HIE had worse pH (difference 0.3 pts, p < 0.01), base deficit (difference of 13 points, p <0.01), 1- and 5-minute Apgars (difference of 6 and 5 points respectively, both with p < 0.01), and SNAPPE scores (difference of 16.5 points, p < 0.01) than those with CHD.

The abnormalities seen on MRI are described in detail in S1 and S2 Tables. 66.7% of the HIE cohort had normal MRIs. Of the 20 HIE subjects with abnormal MRIs, findings were restricted to focal abnormality in a watershed region in 8 (40%), abnormal signal in anterior or

**Table 2. Clinical characteristics.**

|  | HIE | | | CHD | | | HIE vs. CHD |
|---|---|---|---|---|---|---|---|
|  | **n = 60** | | | **n = 35** | | | |
|  | **Normal MRI** | **Injury on MRI** |  | **Normal MRI** | **Injury on MRI** |  |  |
|  | **40 (66.7%)** | **20 (33.3%)** |  | **22 (62.9%)** | **13 (37.1%)** |  |  |
|  |  |  |  | **(SV 13, TGA 9)** | **(SV 7, TGA 6)** |  |  |
| **Characteristic–Median (IQR)** |  |  | **p** |  |  | **p** | **Medians, p** |
| Injury on MRI | - | - | - | - | - | - | 33.3% vs. 37.1%, 0.82 |
| pH at Delivery | 7.00 (6.89–7.07) | 6.96 (6.82–7.10) | 0.40 | 7.29 (7.23–7.34) | 7.25 (7.19–7.31) | 0.22 | **6.98 vs. 7.28, <0.01** |
| Base Deficit at Delivery | -15 (-19 - -12) | -17 (-26 - -13) | 0.26 | -3 (-4 - -2) | -5 (-6 - -2) | 0.36 | **-16 vs -3, <0.01** |
| Apgar at 1 Minute | 2.0 (1–3) | 2.0 (1–3) | 0.89 | 8.0 (7–8) | 8.0 (7–8) | 0.89 | **2 vs. 8, <0.01** |
| Apgar at 5 Minutes | 4.0 (3–6) | 4.5 (2–6.5) | 0.86 | 9.0 (9–9) | 9.0 (9–9) | 1.00 | **4 vs. 9, <0.01** |
| Apgar at 10 Minutes | 6.0 (4–7) | 7.0 (5–8) | 0.23 | 7.0 (7–7) | 8.0 (7–9) | 1.00 | **6 vs 7, 0.12** |
|  | *n = 35* | *n = 20* |  | *n = 2* | *n = 2* |  |  |
| SNAPPE | 29 (25–34) | 31 (27–34) | 0.99 | 14 (5–16) | 15 (5–23) | 0.71 | **31.0 vs. 14.5, <0.01** |
|  | *n = 15* | *n = 10* |  | *n = 15* | *n = 11* |  |  |
| Seizures Present DOL 1–3, n (%) | 4 (11.4%) | 2 (18.8%) | 1 | - | - | - | - |
|  | *n = 35* | *n = 14* |  |  |  |  |  |

HIE, hypoxic-ischemic encephalopathy; CHD, congenital heart disease; SV, single ventricle; TGA, transposition of the great arteries; IQR, interquartile range; SNAPPE, score for neonatal acute physiology—perinatal extension; DOL, day of life.

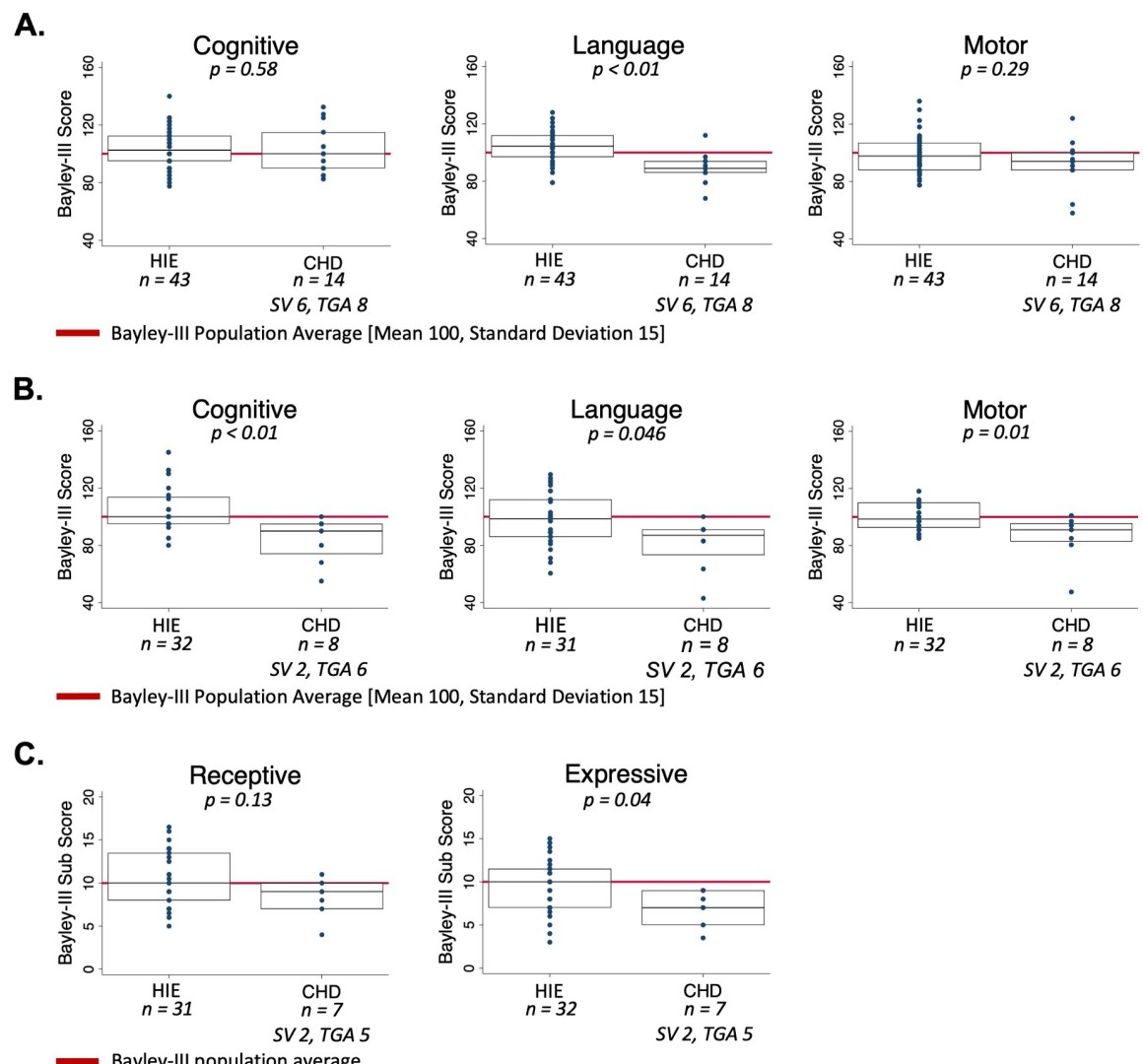

**Fig 3. Neurodevelopmental outcomes in hypoxic-ischemic encephalopathy and congenital heart disease.** (A) Bayley-III scores at 12–18 months of age for cognitive, language, and motor domains. (B) Bayley-III scores at 30 months of age for cognitive, language, and motor domains. (C) Bayley-III sub scores for receptive and expressive language at 30 months of age.

posterior watershed white matter in 2 (10%), abnormal signal in white matter and cortex in an anterior or posterior watershed zone in 4 (20%), and abnormal signal in both anterior and posterior watershed regions in 2 (10%). Only 4 (20%) had basal ganglia injury—2 with abnormal signal restricted to the thalamus and 2 with abnormal signal in the thalamus and lentiform nucleus. 62.9% of the CHD cohort had normal MRIs. Of the 13 CHD subjects with abnormal MRIs, 3 (8.6%) had mild white matter injury, 6 (17.1%) had moderate to severe white matter injury, and 4 (11.4%) had stroke.

## Neurodevelopmental outcomes

At 12–18 months, Bayley-III testing was available for 42 HIE patients and 14 CHD patients (Fig 3A). In all Bayley-III domains, the HIE cohort scored within or above the normal range. CHD patients had decreased language scores compared the HIE group (HIE median

score = 104.5, interquartile range (IQR) 97–112; CHD median score = 89, IQR 86–94, p < 0.01). No differences between HIE and CHD cohorts were seen in the cognitive or motor domains at this timepoint. At this timepoint, 92.9% of CHD patients scored within or above the normal population range for the cognitive domain, 78.6% scored within or above the normal range for the language domain, and 85.7% scored within or above the normal range for the motor domain. Within each cohort, we did not observe differences in outcomes between those with and without injury on MRI (S3 Table).

At 30 months, Bayley-III testing was available for 31 HIE patients and 8 CHD patients (Fig 3B). In all Bayley-III domains, the HIE cohort scored within or above the normal range. At this timepoint in the CHD cohort, 62.5% scored the normal range for the cognitive domain, 50% scored within or above the normal range for the language domain, and 75% scored within or above the normal range for the motor domain. In the cognitive domain, the HIE median score was 100 (IQR 95–114) and the CHD median score was 90 (IQR 74–95), p < 0.01. In the language domain, the HIE median score was 98.5 (IQR 86–112) and the CHD median score was 87 (IQR 73.5–91), p = 0.046. In the motor domain, the HIE median score was 98.5 (IQR 92.5–110) and the CHD median score was 91 (IQR 83–95.5), p = 0.01. Within each cohort, we did not observe differences in outcomes in any domain between those with and without normal imaging or those with SV versus those with TGA (S3 Table). Expressive and receptive language sub scores were analyzed individually (Fig 3C). CHD patients had lower expressive language but not receptive language sub scores (HIE median expressive language Subscore 10, IQR 7–11.5l CHD Subscore 7 IQR 5–9, p = 0.04).

## Graph metrics

Global efficiency, a marker of overall brain integration, was reduced in CHD compared to HIE (p = 0.03) (Fig 4). No differences were observed in transitivity, modularity, and small-world-ness. No differences in graph metrics were observed within each cohort when comparing patients with MRI injury to those without injury or when comparing SV to TGA (summarized in S4 Table).

## Univariate analysis—Predictors of functional outcomes

A univariate analysis was performed assessing for neonatal factors associated with early childhood neurodevelopment via linear regression (Tables 3 and 4). Factors previously described to be associated with outcome, such as injury on MRI, sex, race, maternal education, preoperative level of illness, birth weight, and gestational age at birth were included. Delivery pH, base deficit, and Apgars were not included because they were inclusion criteria to the BAMRI.

12–18 month language outcomes were associated with maternal level of education, sex, and cohort grouping. 12–18 month motor outcomes were associated with by sex (Table 3). Graph metrics were not associated with outcome in any domain at this timepoint (Table 3).

At 30 months, cognitive outcomes were associated with cohort group (CHD with lower scores), sex (males with lower scores), maternal education, and gestational age at birth and at scan (Table 4). Language outcomes were associated with cohort (CHD with lower scores) and gestational age at scan. Motor outcomes were associated with cohort, sex (CHD with lower scores), global efficiency, and transitivity (Table 4).

## Multivariate analysis—Network predictors of functional outcome

Structural connectivity was associated with 30 month motor outcomes on univariate analysis. We assessed each graph metric in a separate multivariate model because graph metrics are not mathematically independent (Table 5). Neonatal global efficiency was positively correlated

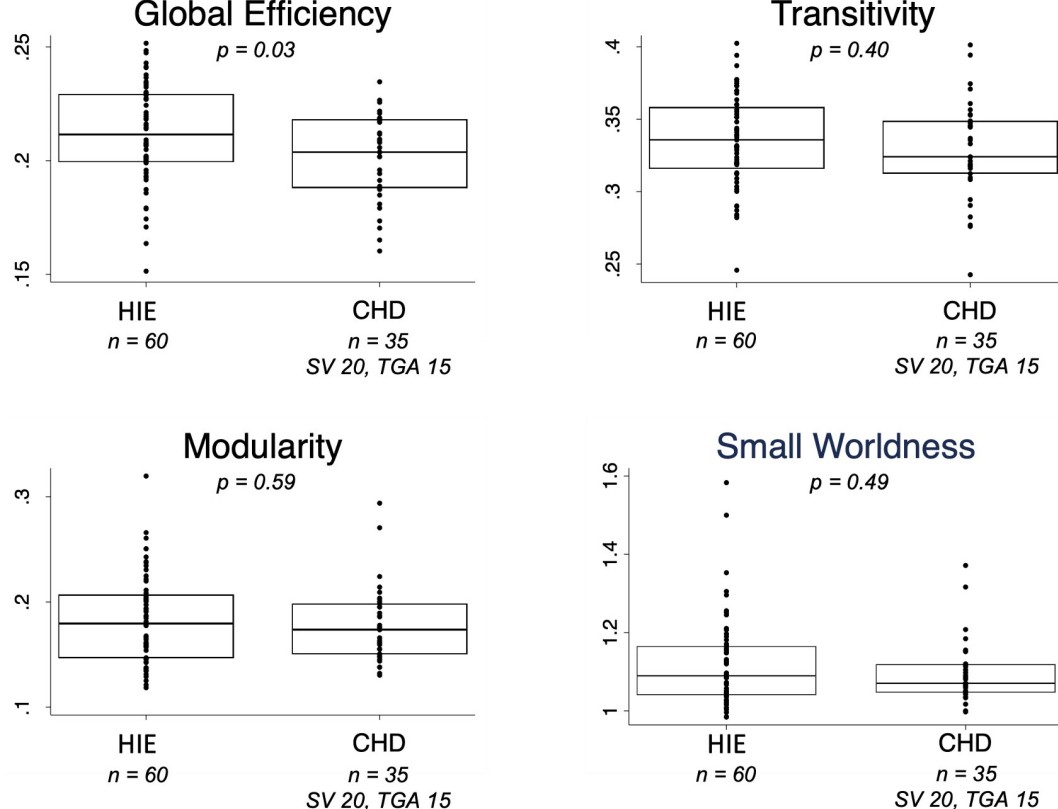

**Fig 4. Global graph metrics in hypoxic-ischemic encephalopathy and congenital heart disease.** Boxplots and overlying scatter plots depict the medians, interquartile ranges, and individual data points for the following graph metrics: global efficiency, transitivity, modularity, and small-worldness.

with 30 month motor outcome. Global efficiency is the inverse average shortest path length and therefore was log transformed. For each unit increase in the log of global efficiency, the motor score was on average 43 points higher, CI 6.5–80.8, p = 0.02. Transitivity was positively correlated as well, albeit at a trend level (p = 0.07). Because structural connectivity metrics were not associated with outcomes at 12–18 months of life on univariate analysis, a multivariate analysis at this time point was not performed.

### Hypothesis-based analysis of language pathways

The CHD cohort had pronounced delays in the language domain at both 12–18 and 30 months. From whole connectomes, individual pathways known to be involved in adult language function were identified and assessed. On univariate analysis, averaged FA along the left SLF tract 2, which connects the inferior parietal to the frontal lobe was associated with expressive language sub scores at 30 months (p = 0.05) (Table 6). Similarly, averaged FA along the left SLF tract 3, which connects the supramarginal gyrus to the frontal lobe, was associated with composite language outcomes at 12–18 months (p = 0.05).

### Hypothesis-free whole connectome edge analysis—CHD vs. HIE

Whole connectome analysis was performed using network based statistics [30]. After false discovery rate correction and controlling for gestational age at MRI, 18 pathways were

**Table 3. Univariate analysis of 12-18-month neurodevelopmental outcomes.**

| Clinical factors | Cognitive | | Language | | Motor | |
|---|---|---|---|---|---|---|
| | Coeff. (95% CI) | p | Coeff. (95% CI) | p | Coeff. (95% CI) | p |
| Cohort (ref: HIE) | -1.7 (-10.6–7.2) | 0.70 | **-16.1 (-23 - -9.2)** | **<0.01** | -6.6 (-15.6–2.4) | 0.15 |
| Sex (ref: male) | 5.0 (-2.7–12.7) | 0.20 | **9.1 (2.4–15.8)** | **0.01** | **8.4 (0.7–16)** | **0.03** |
| MRI abnormal | -2.2 (-10.6–6.1) | 0.59 | 5.5 (-2–13.1) | 0.15 | -1.7 (-10.3–6.9) | 0.69 |
| Race (ref: white) | Prob > F = 0.47 | | Prob > F = 0.98 | | Prob > F = 0.49 | |
| Asian | -9.1 (-24.6–6.3) | 0.24 | -3.7 (-18.4–11) | 0.62 | 1.3 (-14.6–17.2) | 0.87 |
| Pacific islander | 0.9 (-20.2–22) | 0.93 | -6.5 (-26.6–13.5) | 0.52 | -16.6 (-38.3–5.1) | 0.13 |
| Black | 3.4 (-17.7–24.5) | 0.75 | 0.5 (-19.6–20.5) | 0.96 | 8.2 (-13.5–29.9) | 0.45 |
| Latin-x | 0.5 (-9–10) | 0.92 | 0.2 (-8.8–9.3) | 0.96 | -3.6 (-13.4–6.2) | 0.46 |
| Other | -9.8 (-21.4–1.9) | 0.10 | -0.3 (-11.3–10.8) | 0.96 | -6.4 (-18.3–5.6) | 0.29 |
| Maternal education (ref: graduate school degree) | Prob > F = 0.84 | | **Prob > F: <0.01** | | Prob > F = 0.15 | |
| Partial elementary school | -1.6 (-25.9–22.7) | 0.90 | -5.5 (-21.7–10.8) | 0.50 | -0.0 (-23–22.9) | 1 |
| Partial middle school | -4.1 (-37.1–28.9) | 0.80 | -27.0 (-49.1 - -4.8) | 0.02 | 2.5 (-28.7–33.6) | 0.87 |
| Partial high school | -9.9 (-30.5–10.6) | 0.34 | -19.6 (-33.4 - -5.8) | 0.01 | 6.5 (-13–25.9) | 0.51 |
| High school graduate | -9.3 (-22.2–3.7) | 0.15 | -17.1 (-25.8 - -8.4) | <0.01 | -7.4 (-19.6–4.8) | 0.23 |
| Partial college | -8.5 (-26.9–10) | 0.36 | 4.9 (-7.4–17.3) | 0.43 | -11.0 (-28.5–6.4) | 0.21 |
| College graduate | -4.5 (-17.4–8.5) | 0.49 | -0.6 (-9.3–8.1) | 0.89 | 8.1 (-4.1–20.3) | 0.19 |
| SNAPPE | -0.2 (-0.7–0.2) | 0.25 | 0.1 (-0.4–0.6) | 0.67 | -0.0 (-0.5–0.5) | 0.96 |
| Gestational age at MRI (weeks) | 1.8 (-0.6–4.3) | 0.14 | -0.6 (-2.9–1.7) | 0.58 | 1.2 (-1.3–3.8) | 0.34 |
| Gestational age at birth (weeks) | 1.7 (-0.8–4.2) | 0.19 | -0.8 (-3.1–1.5) | 0.48 | 1.1 (-1.5–3.7) | 0.41 |
| Birth weight (kg) | 4.4 (-3–11.7) | 0.24 | -0.6 (-7.5–6.2) | 0.85 | -4.1 (-11.6–3.5) | 0.28 |
| **Graph Metrics** | | | | | | |
| Global efficiency | 8.3 (-25.9–42.5) | 0.63 | -9.6 (-40.9–21.7) | 0.54 | 12.7 (-22.3–47.7) | 0.47 |
| Modularity | -10.2 (-28.6–8.1) | 0.27 | 2.8 (-14.1–19.8) | 0.74 | 0.4 (-18.6–19.5) | 0.96 |
| Transitivity | -3.1 (-39.4–33.1) | 0.86 | 3.9 (-29.4–37.1) | 0.82 | 10.4 (-26.7–47.6) | 0.58 |
| Small-worldness | -26.2 (-77.8–25.4) | 0.31 | 12.2 (-35.4–59.8) | 0.61 | 3.6 (-49.9–57) | 0.89 |

SNAPPE, score for neonatal acute physiology—perinatal extension. Coefficients and p-values are calculated using univariate linear regression. Graph metrics are log transformed.

hypoconnected in the CHD cohort as compared to the HIE cohort (Fig 5). Only one pathway in the HIE cohort were hypoconnected compared to CHD. The most commonly affected pathways involve left sided occipital and temporal regions.

## Discussion

In this study of two well characterized cohorts at risk for neonatal brain injury, we found that CHD newborns had significantly worse language function at 12–18 months, and worse cognitive, language and motor function at 30 months compared with HIE newborns treated with hypothermia therapy. Structural connectomes derived from diffusion MRI of CHD newborns showed a pattern consistent with overall delayed development and hypoconnectivity. Global efficiency, a measure of integration, was significantly lower in CHD compared with HIE. Other topological measures of structural connections (modularity, transitivity, small-worldness) were similar in both cohorts. After correcting for significant covariates known to affect neurodevelopmental outcomes, we found that global efficiency was highly associated with motor outcomes at 30 months in both groups. In light of the consistent and significantly reduced language function in the CHD group, we explored language pathways using both

**Table 4. Univariate analysis of 30 month neurodevelopmental outcomes.**

| | Cognitive | | Language | | Motor | |
|---|---|---|---|---|---|---|
| **Clinical factors** | **Coeff. (95% CI)** | **p** | **Coeff. (95% CI)** | **p** | **Coeff. (95% CI)** | **p** |
| Cohort (ref: HIE) | **-21.5 (-34.9 - -8.1)** | **<0.01** | **-17.9 (-32.5 - -3.2)** | **0.02** | **-14.6 (-23.7 - -5.5)** | **<0.01** |
| Sex (ref: male) | **12.3 (0.8–23.7)** | **0.04** | 8.9 (-3.6–21.3) | 0.16 | **8.2 (0.4–16)** | **0.04** |
| MRI abnormal | 7.4 (-5.2–20.1) | 0.24 | 6.5 (-7.2–20.2) | 0.34 | 2.9 (-5.9–11.6) | 0.51 |
| Race (ref: white) | Prob > F = 0.79 | | Prob > F = 0.80 | | Prob > F = 0.45 | |
| Asian | -1.7 (-26.5–23.1) | 0.89 | -4.6 (-30.7–21.5) | 0.72 | -0.6 (-17–15.8) | 0.94 |
| Pacific islander | -11.7 (-52.2–28.8) | 0.56 | -10.5 (-52.9–31.7) | 0.62 | 10.4 (-16.3–37.1) | 0.43 |
| Black | -6.7 (-36.2–22.8) | 0.65 | -8.0 (-39–23) | 0.60 | 6.4 (-13.1–25.9) | 0.51 |
| Latin-x | -9.4 (-24.6–5.8) | 0.22 | -11.0 (-27.1–5.2) | 0.18 | -2.9 (-12.9–7.1) | 0.56 |
| Other | -10.3 (-28.2–7.7) | 0.25 | -1.0 (-19.9–18) | 0.92 | -9.7 (-21.5–2.2) | 0.11 |
| Maternal education (ref: graduate school degree) | **Prob > F = 0.03** | | Prob > F = 0.09 | | Prob > F = 0.40 | |
| Partial high school | -17.9 (-58.9–23.2) | 0.38 | -17.4 (-59–24.1) | 0.40 | -10.7 (-41–19.6) | 0.47 |
| High school graduate | -25.1 (-43.6 - -6.5) | 0.01 | -24.6 (-43.3 - -5.8) | 0.01 | -10.0 (-23.7–3.7) | 0.15 |
| Partial college | -12.9 (-35.3–9.6) | 0.25 | -11.9 (-34.7–10.8) | 0.29 | -5.1 (-21.7–11.5) | 0.53 |
| College graduate | 2.5 (-18–23) | 0.80 | -4.3 (-25.1–16.5) | 0.67 | 1.9 (-13.3–17) | 0.80 |
| SNAPPE | 0.2 (-0.6–0.9) | 0.61 | 0.4 (-0.4–1.3) | 0.30 | 0.1 (-0.5–0.7) | 0.77 |
| Gestational age at MRI (weeks) | **3.8 (0.2–7.4)** | **0.04** | 3.5 (-0.3–7.3) | 0.07 | 1.7 (-.8–4.2) | 0.18 |
| Gestational age at birth (weeks) | **3.6 (-0.1–7.2)** | **0.05** | 3.1 (-0.8–7) | 0.11 | 1.8 (-0.8–4.3) | 0.17 |
| Birth weight (kg) | 8.0 (-3–19) | 0.15 | 3.2 (-8.6–15) | 0.59 | 2.7 (-5–10.4) | 0.48 |
| **Graph metrics** | | | | | | |
| Global efficiency | 28.7 (-23.2–80.5) | 0.27 | 14.1 (-41–69.2) | 0.61 | **48.6 (16.4–80.7)** | **<0.01** |
| Modularity | -15.7 (-43.6–12.3) | 0.26 | -9.1 (-39.9–21.8) | 0.56 | -16.1 (-34.7–2.5) | 0.09 |
| Transitivity | 30.6 (-26.5–87.7) | 0.28 | 3.1 (-64.7–71) | 0.93 | **45.8 (9.3–82.4)** | **0.02** |
| Small worldness | -13.5 (-90.7–63.7) | 0.73 | 0.2 (-88–88.4) | 1 | -34.3 (-85.7–17.2) | 0.19 |

SNAPPE, score for neonatal acute physiology—perinatal extension. Coefficients and p-values are calculated using univariate linear regression. Graph metrics are log transformed.

hypothesis-based and data-driven approaches. We found that the SLF 3 was negatively correlated with overall language function at 12–18 months and SLF 2 was negatively correlated with expressive language at 30 months. After controlling for multiple comparisons and gestational age at scan, we found a number of white matter tracts and regions of interest to be hypoconnected in the CHD group compared with HIE. Among these are regions known to be associated with language and visual function suggesting specific anatomical loci for deficits observed in children with CHD.

**Table 5. Multivariate analyses of motor outcomes.**

| Variable | Global efficiency | | Modularity | | Transitivity | |
|---|---|---|---|---|---|---|
| | Prob > F: <0.01 | | Prob > F: <0.01 | | Prob > F: <0.01 | |
| | **Coeff. (95% CI)** | **p** | **Coeff. (95% CI)** | **p** | **Coeff. (95% CI)** | **p** |
| Graph metric | **43.7 (6.5–80.8)** | **0.02** | -14.2 (-31.1–2.7) | 0.10 | 31.4 (-2.8–65.6) | 0.07 |
| Cohort (ref: HIE) | -11.1 (-19.7 - -2.4) | 0.01 | -12.7 (-21.4 - -4) | 0.01 | -13.0 (-21.6 - -4.5) | <0.01 |
| Sex (ref: male) | 7.4 (0.2–14.7) | 0.04 | 7.7 (0.1–15.3) | 0.05 | 4.9 (-2.6–12.5) | 0.20 |
| Gestational age at MRI (weeks) | -1.1 (-3.9–1.7) | 0.44 | 0.4 (-2–2.8) | 0.73 | 0.6 (-1.7–3) | 0.57 |

Model probability, coefficients, and p-values calculated using multivariate linear regression.

**Table 6. Language outcomes and neural pathways.**

| Pathway | Language (12–18 Months) | | Language (30 Months) | | Expressive Language (30 Months) | | Receptive Language (30 Months) | |
|---|---|---|---|---|---|---|---|---|
| | Coeff. (95% CI) | p | Coeff. (95% CI) | p | Coeff. (95% CI) | p | Coeff. (95% CI) | p |
| SLF 1 | 5.7 (-11–22.3) | 0.50 | 2.0 (-28.3–32.4) | 0.89 | -0.2 (-5–4.6) | 0.94 | -2.7 (-7.4–2) | 0.25 |
| SLF 2 | 4.0 (-16.4–24.3) | 0.70 | -19.1 (-53.1–15) | 0.26 | **-5.2 (-10.4–0)** | **0.05** | -4.3 (-9.6–1) | 0.11 |
| SLF 3 | **-8.9 (-17.8 - -0.1)** | **0.05** | 4.6 (-10.4–19.5) | 0.54 | 0.5 (-1.9–2.8) | 0.69 | -0.3 (-2.7–2.1) | 0.81 |
| ILF | -2.3 (-7.7–3.1) | 0.40 | 1.0 (-8.5–10.5) | 0.83 | 0.7 (-0.8–2.2) | 0.37 | 0.5 (-1–2) | 0.53 |
| Arcuate | -5.3 (-20.2–9.6) | 0.48 | -4.8 (-30.5–21) | 0.71 | 1.6 (-2.6–5.9) | 0.43 | 1.6 (-2.6–5.9) | 0.44 |

SLF, superior longitudinal fasciculus; ILF inferior longitudinal fasciculus. Coefficients and p-values reported from univariate linear regression analysis

The most prominent difference in white matter topology in CHD compared to HIE was reduced global efficiency, a parameter measuring overall brain integration. Other studies have noted similar reductions in global efficiency in neonates with CHD compared to healthy controls [15, 16] and this persists into adolescence, albeit to a lesser extent [31]. After controlling for cost, Schmithortz et al found increased small-worldness, a metric related to the balance between network integration and segregation and hypothesized this was due to adaptive energy-efficient fetal rewiring in response to unmet metabolic demands of the developing

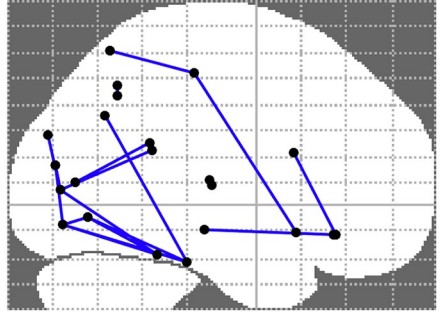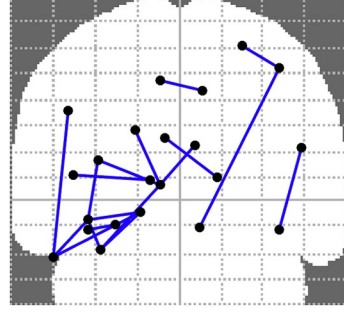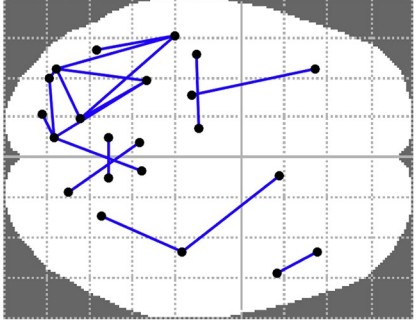

| Pathway | Test statistic | Region of interest | Frequency of involvement |
|---|---|---|---|
| Left Calcarine to Left Fusiform | 4.62 | Left Calcarine | 4 |
| Right Olfactory to Right Postcentral | 4.39 | Left Inferior Occipital | 4 |
| Left Calcarine to Left Middle Occipital | 4.11 | Left Fusiform | 3 |
| Left Precuneus to Right Precuneus | 4.05 | Left Lingual | 3 |
| Left Lingual to Left Inferior Temporal | 3.93 | Left Inferior Temporal | 3 |
| Left Thalamus to Left Heschl | 3.87 | | |
| Left Calcarine to Left Superior Occipital | 3.83 | | |
| Left Middle Occipital to Left Inferior Occipital | 3.81 | | |
| Left Angular to Left Inferior Temporal | 3.80 | | |
| Left Inferior Occipital to Left Inferior Temporal | 3.75 | | |

**Fig 5. Hypoconnected pathways and associated regions of interest in congenital heart disease.** After controlling for a false discovery rate and corrected gestational age at MRI, 18 weaker connections involving 22 ROIs were identified in CHD. The top 10 hypoconnections are listed, along with associated their test statistics. The top 6 ROIs most frequently involved in the hypoconnected pathways are also listed. A pictorial representation of the hypoconnections and associated ROIs is provided.

brain in CHD [16]. We did not observe this, however we did not control for cost or strength. We chose not to cost control in our study because the independence of total regional cost (or total regional strength in our non-binary networks) and network organization is not established. Feldmann et al also noted reduced global efficiency on preoperative CHD scans compared to normal controls [15]. They also found significantly higher local efficiency and transitivity, measures of network segregation and concluded that this represents delayed brain maturation in CHD neonates.

The links between structural connectivity and functional outcome are poorly understood. To our knowledge, no study has analyzed the association of network topological parameters on neonatal imaging with subsequent neurodevelopmental outcomes in childhood. Several studies have examined brain network organization in childhood in relation to concurrent functional outcomes. Adolescents with TGA continue to show a trend towards decreased global efficiency [31]. Differences in network topology in TGA adolescents (decreased global efficiency, increased modularity and small-worldness) appeared to mediate worse cognitive performance across multiple domains including overall intelligence, memory, executive and visual-spatial functions [31]. In a separate study, teenage subjects with TGA exhibited worse performance on testing for attention deficit hyperactivity disorder (ADHD) compared with controls and performance was mediated by changes in overall network connectivity [32].

Very few links have been established between findings on neonatal imaging in general and later functional outcomes in childhood. We recently reported that moderate to severe focal white matter injury (WMI) was associated with motor function at 30 months but not earlier [33]. Similarly, in the present study we found that neonatal global efficiency and, to a lesser extent, transitivity predicted motor outcomes at 30 months of life but not earlier at 12–18 months. Overall, we did not see a relationship between global efficiency and brain injury on MRI, but we did observe a trend-level association in the HIE cohort between reduced global efficiency and presence of injury on imaging. These results imply that brain injury and topological connectivity independently predict motor outcome at 30 months. While earlier motor assessment is important to facilitate timely intervention, imaging appears to predict later and not early motor outcomes.

One striking finding in our study is how much worse the CHD group performed compared to the HIE group. The CHD cohort had significantly worse language function at 12–18 months and worse cognitive, language and motor function at 30 months. This finding was not due to inclusion of newborns with mild encephalopathy. The HIE group was selected based upon standard clinical definitions for moderate to severe neonatal encephalopathy (e.g., severe acidosis, low 5 minute Apgar scores, moderate encephalopathy). Despite moderate to severe encephalopathy, the majority of HIE subjects (65%) had normal MRIs, consistent with the therapeutic benefit of hypothermia therapy [34]. Outcomes in the HIE group were within the normal range at both 12–18 and 30 months. The CHD group also had a majority with normal imaging (63%) on the preoperative scan. Despite this, outcomes for the CHD were significantly worse at 12–18 months for language and were worse at 30 months for cognitive, language and motor. While 30% of those with CHD will acquire new brain injury postoperatively [21], and those with SV will have continued cyanosis for years to come, group differences may also be explained by the timing and mechanism of brain injury.

Disruption of fetal circulation with TGA and HLHS exists throughout gestation [35]. Beginning in the third trimester, spontaneous patterned activity arises at many levels of the developing nervous system [8]. Activity in the forms of spontaneous waves serves to provide patterned input to developing circuits [8] and drives many aspects of activity dependent neuronal development [36]. Neuronal activity is energetically costly, and to meet increased brain oxygen consumption, the proportion of total cardiac output to the brain increases markedly in the third

trimester in humans [37]. Diminished oxygen/nutrient delivery and/or flow with CHD leads to a decline in brain growth and maturation during the third trimester [38, 39] and CHD newborns born at term have delayed brain development [40]. Although it has not been studied directly, we would predict that spontaneous brain activity is also reduced in the fetus with CHD as a consequence of chronically decreased oxygen delivery. The protracted nature of disrupted fetal circulation and delayed brain development may lead to more widespread dysmaturation of developing cortical circuits in CHD. In comparison, HIE has mostly normal third trimester brain development until the onset of subacute or acute hypoxia ischemia just prior to or at delivery. While hypoxic ischemia can result in devastating destructive brain injury in selected cases, the condition is known to result in a range of injury severity. Furthermore, recent widespread adoption of hypothermia therapy for moderate to severe encephalopathy has reduced overt brain injury on MRI [34] and improved outcomes [41].

Normal brain network connectivity develops from a state of small-world segregation characterized by low global efficiency to increasing integration with reorganization of connections and increasing global efficiency [42]. This reorganization is also accompanied by an increase in connection strength. In that light, reduced global efficiency in CHD may represent delayed development. When whole connectomes are compared between CHD and HIE, with false discovery correction 18 hypoconnected pathways were identified in the CHD group and only one pathway had increased connection strength, a finding also suggestive of delayed development. On univariate and multivariate analysis, global efficiency in the neonatal period was found to be associated with motor outcomes at 30 months of life. Efficient networks are metabolically demanding, and thus the importance of neonatal global efficiency in later motor development may indicate a reliance on adaptive network redundancy that likely undergoes later pruning and refinement due to the diffuse nature of cortical requirements to enact complex motor sequences. Insufficient substrate for later remodeling may result in impaired motor development. Alternatively, the metabolic requirements of early brain network efficiency and normal motor development may be inadequately met in those with congenital cyanotic heart disease. If this is true, we should see more rapid improvements in both global efficiency and motor outcomes in the TGA subgroup as compared to those with SV, but we do not as yet have serial imaging or sufficient sample size to test this hypothesis. DTI imaging at sequential time points in the CHD cohort would elucidate this further. For example, this would be supported if, postoperatively, the rate of rise in efficiency in those with TGA (as opposed to SV) were similar to non-cardiac infants and greater in comparison to newborns with SV, similar to findings we have previously reported of improved postoperative brain growth in TGA [43].

Of all three domains tested by the Bayley III, the most pronounced delays in CHD were seen in language development. Impaired communication has been reported in children with critical CHD, although receptive language scores are in the average range [44]. Thus, delays in language development appear to be driven more so by expressive language [45]. We assessed specific pathways known to be involved in language function including the AF, ILF, and three tracts of the SLF [46]. The superior longitudinal fasciculus, a large neural bundle made up of several sub-tracts, connects the parietal and frontal cortex and is implicated in language production, motor behavior, and spatial attention [47–49]. The inferior longitudinal fasciculus connects regions of the temporal and occipital cortexes and is implicated in visual perception, reading, semantical processing, and some autism spectrum disorders [50, 51]. The arcuate fasciculus connects regions of the temporal and frontal cortexes and is implicated in phonological processing in younger children and reading ability in older children [29, 52]. Only SLF tract 3 was associated with composite language scores at 12–18 months in the CHD group, with an inverse relationship. At 30 months, SLF tract 2 was inversely associated with expressive language performance. Other pathways involved in top down and bottom up language processing

including the ILF and arcuate fasciculus were not associated with language function. These results suggest that early language delays in CHD may be more related to motor deficits and expressive language than with receptive processing and comprehension. Undoubtedly, memory and attention also play a role in language function. Similar to our findings, studies in children with both autism spectrum disorders and language delays have also seen negative correlations between SLF connectivity and language ability [53].

When whole connectomes were compared using a false discovery rate correction, among the regions most commonly hypoconnected in neonatal CHD patients was the left fusiform gyrus, a brain region that has been functionally associated with auditory language and semantic processing in older children and adults [54]. Other notable hypoconnected areas/pathways include the left calcarine to middle occipital cortex pathway, a pathway involved in visual spatial processing and visual motor integration, and the left interior temporal gyrus which is a region with noted deficiencies after neonatal CHD repair [55, 56].

Our study has a number of limitations including lack of a contemporary normal comparison group, overall sample size, especially for CHD subtypes and associated limited long term follow-up for subgroups. Despite these limitations, this study is the first that we know of to associate neonatal structural connectivity with functional outcomes in childhood. The strongest associations are noted for motor outcomes, a finding we have reported for moderate to severe white matter injury. It is quite likely that specific functional impairments will become more apparent in both HIE and CHD as children grow and are presented with more challenging cognitive tasks in school. Despite scores in the normal range on early testing, both groups are known to require substantial educational services and have increased rates of disorders of attention at school age [3, 44, 57]. In future studies, serial MRIs and neurodevelopmental data from both CHD and a normal cohort of neonates would validate our findings and shed additional light on how the brain network patterns grow and develop in the setting of genetic and metabolic differences in heart disease.

## Conclusions

CHD and HIE are associated with high risk of neonatal brain injury and altered development. Despite substantial injury leading to encephalopathy at birth, HIE newborns perform significantly better than CHD newborns at 12–18 and 30 months, perhaps due to efficacy of hypothermia treatment. Alternatively, the protracted nature of altered fetal circulation may contribute to more profound delay and dysmaturation of brain networks in CHD. For both groups neonatal global efficiency predicts motor function at 30 months even after controlling for clinical and sociodemographic factors and thus may be a useful imaging biomarker for children at need of close follow-up and early intervention. Finally, hypothesis-based and data-driven whole brain analyses identified specific pathways and areas that may engender neurodevelopmental deficits including problems with expressive language and visual spatial processing.

## Supporting information

**S1 Table. MRI injury patterns in hypoxic-ischemic encephalopathy.**
(DOCX)

**S2 Table. MRI injury patterns in congenital heart disease.**
(DOCX)

**S3 Table.** A. Neurodevelopmental outcomes in those with and without injury on MRI. B. Neurodevelopmental outcomes in those with SV and TGA.
(DOCX)

**S4 Table.** A. Graphics metrics in those with and without injury on MRI. B. Graphics metrics in single ventricle and transposition of the great arteries.
(DOCX)

## Author Contributions

**Conceptualization:** Alice Ramirez, Shabnam Peyvandi, Olga Tymofiyeva, Patrick S. McQuillen.

**Data curation:** Alice Ramirez, Shabnam Peyvandi, Stephany Cox, Dawn Gano, Duan Xu, Olga Tymofiyeva, Patrick S. McQuillen.

**Formal analysis:** Alice Ramirez, Shabnam Peyvandi, Olga Tymofiyeva, Patrick S. McQuillen.

**Funding acquisition:** Shabnam Peyvandi, Patrick S. McQuillen.

**Investigation:** Alice Ramirez, Dawn Gano, Duan Xu, Olga Tymofiyeva, Patrick S. McQuillen.

**Methodology:** Alice Ramirez, Shabnam Peyvandi, Stephany Cox, Dawn Gano, Duan Xu, Olga Tymofiyeva, Patrick S. McQuillen.

**Project administration:** Patrick S. McQuillen.

**Resources:** Duan Xu, Olga Tymofiyeva, Patrick S. McQuillen.

**Software:** Duan Xu, Olga Tymofiyeva.

**Supervision:** Shabnam Peyvandi, Stephany Cox, Dawn Gano, Olga Tymofiyeva, Patrick S. McQuillen.

**Validation:** Shabnam Peyvandi, Stephany Cox, Dawn Gano, Duan Xu, Olga Tymofiyeva, Patrick S. McQuillen.

**Visualization:** Alice Ramirez, Olga Tymofiyeva.

**Writing – original draft:** Alice Ramirez.

**Writing – review & editing:** Alice Ramirez, Shabnam Peyvandi, Stephany Cox, Dawn Gano, Duan Xu, Olga Tymofiyeva, Patrick S. McQuillen.

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
