## [Decision Letter · Decision Letter 0]

13 Sep 2021

PONE-D-21-21117Neonatal brain injury influences structural connectivity and childhood functional outcomesPLOS ONE

Dear Dr. McQuillen,

Thank you for submitting your manuscript to PLOS ONE. After careful consideration, we feel that it has merit but does not fully meet PLOS ONE’s publication criteria as it currently stands. Therefore, we invite you to submit a revised version of the manuscript that addresses the points raised during the review process.

We look forward to receiving your revised manuscript.

Kind regards,

Emma Duerden

Academic Editor

PLOS ONE

Journal Requirements:

3. We note that you have included the phrase “data not shown” in your manuscript. Unfortunately, this does not meet our data sharing requirements. PLOS does not permit references to inaccessible data. We require that authors provide all relevant data within the paper, Supporting Information files, or in an acceptable, public repository. Please add a citation to support this phrase or upload the data that corresponds with these findings to a stable repository (such as Figshare or Dryad) and provide and URLs, DOIs, or accession numbers that may be used to access these data. Or, if the data are not a core part of the research being presented in your study, we ask that you remove the phrase that refers to these data

Reviewers' comments:

Reviewer's Responses to Questions

**Comments to the Author**

1. Is the manuscript technically sound, and do the data support the conclusions?

Reviewer #1: Yes

Reviewer #2: Partly

2. Has the statistical analysis been performed appropriately and rigorously? 

Reviewer #1: Yes

Reviewer #2: I Don't Know

3. Have the authors made all data underlying the findings in their manuscript fully available?

Reviewer #1: Yes

Reviewer #2: Yes

4. Is the manuscript presented in an intelligible fashion and written in standard English?

Reviewer #1: Yes

Reviewer #2: Yes

5. Review Comments to the Author

Reviewer #1: Thank you for the opportunity to review this very interesting manuscript.

This is a well-written manuscript practicing high-quality reporting of research methods and findings overall. I only have a few comments:

Although I found very interesting the choice of comparing these two different clinical cohorts, there is a need for a stronger rationale to do this and considering the lack of comparison to healthy controls.

P.9 line 121, the two newborns who did not receive TH should be excluded from the analyses. I understand that the results may not have been different, but they are just inducing noise considering that they are not meeting the neat inclusion criteria. Particularly the one with mild HIE and considering that none had neurodevelopmental follow-up.

Was CP an exclusion criteria? I was surprised to see no case of suspected CP reported in 42 HIE.

p.17, line 280: Did the CHD patients performed within the normal range for the cognitive and motor domains as well?

p.19 line 315: I recommend being careful when using the term predictor when presenting univariate associations between neurodevelopmental outcome and clinical factors, this study is not powered for this.

p.21 second paragraph: What are the results of the multivariate analysis and outcomes at 12-18months?

p.25 line 432: “transitivity predicted motor outcomes at 2.5 years of life but not earlier at 12-18 months.” Do you mean at 30 months ( not 2.5 years)?

Reviewer #2: First and foremost, I would like to congratulate the authors of this paper on their fine work on a very interesting and complex subject.

The study in my opinion would benefit greatly from a control group, instead of comparing two types of inducers of neonatal brain injuries directly to each other. I see that the groups are sporadic compared to reference data but in general the two groups are being compared to each other.

If it is a possibility, I would recommend doing the comparison with a healthy control group. However, there is some merit in the question of which is worse; CHD or HIE?

Beside this I really enjoyed the article which I found informative and well written. However, much of the content is outside my area of expertise and therefore I do not see myself fit to evaluate the following segments:

• MRI acquisition and processing

• Graph Analysis and the results hereof

6. PLOS authors have the option to publish the peer review history of their article (what does this mean?). If published, this will include your full peer review and any attached files.

Reviewer #1: **Yes: **Marie Brossard-Racine

Reviewer #2: No

---

## [Author Response · Author response to Decision Letter 0]

10 Nov 2021

Dear reviewers, Professor Duerden, and PLOS ONE staff,

We are gratified that both reviewers found our paper to be ‘very interesting’ and ‘well written’. Below we respond to all queries raised by the reviewers.

The major issue common to both reviewers relates to the need for a stronger justification of comparing two clinical cohorts without a cohort of healthy controls. It is unfortunate that the two cohorts compared in this paper do not have contemporary matched normal controls. Mitigating this issue is the fact that all neurodevelopmental outcome assessments used in the paper are based upon population standards, thereby providing a context for each cohort in comparison to the general population. An additional strength of the study is that the imaging methods were identical for each cohort. To justify the comparison, we note that the two conditions (congenital heart disease-CHD and hypoxic ischemic encephalopathy-HIE) are among the highest risk for neonatal brain injury, but that injury occurs with very distinct timing and mechanism in each. In CHD, disrupted fetal circulation leads to impaired brain oxygen and substrate delivery throughout the duration of the 3rd trimester with additional risk for injury before and after surgery. In contrast, HIE results from sub-acute or acute pathology in the days/weeks or hours before birth. Comparing the two populations provides a unique natural ‘experiment’ akin to studies in animal models varying injury timing, mechanism, and severity to determine cell populations and systems with selective vulnerability and mechanisms influencing functional outcomes including repair and plasticity. Our results identify distinct differences between the two cohorts that will be of broad general interest to pediatricians, psychologists, and neuroscientists. The CHD cohort has worse functional outcomes overall, especially language, and a unique pattern of hypoconnectivity in specific language associated circuits. At the same time, a metric of overall network topology (global efficiency) is strongly associated with motor outcomes in both cohorts! 

We have extensively re-written the introduction to emphasize these points and provide a stronger justification for comparing two clinical cohorts. In addition, we have also addressed the journal requirements – we have ensured that our manuscript meets the PLOS One style requirements, we have added supplemental tables to make transparent data that previously was not shown, and we have ensured that the ‘Funding Information’ and ‘Financial Disclosure’ sections match. All other reviewer queries have been addressed with modifications described below.

Reviewer #1

P.9 line 121, the two newborns who did not receive TH should be excluded from the analyses. I understand that the results may not have been different, but they are just inducing noise considering that they are not meeting the neat inclusion criteria. Particularly the one with mild HIE and considering that none had neurodevelopmental follow-up.

These two subjects have been removed and all analyses redone without their data.

Was CP an exclusion criteria? I was surprised to see no case of suspected CP reported in 42 HIE.

Cerebral palsy was not an exclusion criterion. The birth asphyxia cohort was prospectively enrolled and all eligible subjects with suitable neonatal imaging data and neurodevelopmental follow-up were included. We believe the beneficial effects of therapeutic hypothermia account for the reduced rate of cerebral palsy. An alternative explanation is that subjects with severe cerebral palsy did not return for subsequent neurodevelopmental evaluations, however in our cohort we did not see a difference in follow-up rates between those with and without brain injury on imaging to support that hypothesis. 

p.17, line 280: Did the CHD patients performed within the normal range for the cognitive and motor domains as well?

The individual data points for both the CHD and HIE cohorts as well as general population means and standard deviations are provided in figure 3. CHD patients largely performed within one standard deviation of the normal range, and we have added this to the results section to better clarify that finding.

p.19 line 315: I recommend being careful when using the term predictor when presenting univariate associations between neurodevelopmental outcome and clinical factors, this study is not powered for this.

We have replaced the term predictor in the univariate association analyses.

p.21 second paragraph: What are the results of the multivariate analysis and outcomes at 12-18months?

We did not perform a multivariate analysis of outcomes at 12-18 months because the structural connectivity metrics were not found to be associated with outcomes in the univariate analysis at this time point. We have added to the results section to better clarify that finding and decision.

p.25 line 432: “transitivity predicted motor outcomes at 2.5 years of life but not earlier at 12-18 months.” Do you mean at 30 months (not 2.5 years)?

We have replaced 2.5 years with 30 months throughout the manuscript for consistency. 

Reviewer #2

The study in my opinion would benefit greatly from a control group, instead of comparing two types of inducers of neonatal brain injuries directly to each other. I see that the groups are sporadic compared to reference data but in general the two groups are being compared to each other.

If it is a possibility, I would recommend doing the comparison with a healthy control group. However, there is some merit in the question of which is worse; CHD or HIE?

These are valid points and we have endeavored to address them as delineated above.

Thank you for your consideration,

Sincerely,

Patrick McQuillen (corresponding author) and Alice Ramirez (first author)

---

## [Decision Letter · Decision Letter 1]

21 Dec 2021

Neonatal brain injury influences structural connectivity and childhood functional outcomes

PONE-D-21-21117R1

Dear Dr. McQuillen,

We’re pleased to inform you that your manuscript has been judged scientifically suitable for publication and will be formally accepted for publication once it meets all outstanding technical requirements.

Kind regards,

Emma Duerden

Academic Editor

PLOS ONE

Additional Editor Comments (optional):

Reviewers' comments:

Reviewer's Responses to Questions

**Comments to the Author**

1. If the authors have adequately addressed your comments raised in a previous round of review and you feel that this manuscript is now acceptable for publication, you may indicate that here to bypass the “Comments to the Author” section, enter your conflict of interest statement in the “Confidential to Editor” section, and submit your "Accept" recommendation.

Reviewer #1: All comments have been addressed

2. Is the manuscript technically sound, and do the data support the conclusions?

Reviewer #1: Yes

3. Has the statistical analysis been performed appropriately and rigorously? 

Reviewer #1: Yes

4. Have the authors made all data underlying the findings in their manuscript fully available?

Reviewer #1: Yes

5. Is the manuscript presented in an intelligible fashion and written in standard English?

Reviewer #1: Yes

6. Review Comments to the Author

Reviewer #1: (No Response)

7. PLOS authors have the option to publish the peer review history of their article (what does this mean?). If published, this will include your full peer review and any attached files.

Reviewer #1: **Yes: **Marie Brossard-Racine

---

## [Editor Report · Acceptance letter]

27 Dec 2021

PONE-D-21-21117R1 

Neonatal brain injury influences structural connectivity and childhood functional outcomes. 

Dear Dr. McQuillen:

I'm pleased to inform you that your manuscript has been deemed suitable for publication in PLOS ONE. Congratulations! Your manuscript is now with our production department. 

Kind regards, 

on behalf of

Dr. Emma Duerden 

Academic Editor

PLOS ONE